# Enhanced Electrochemical Performance of LiFePO_4_ Originating from the Synergistic Effect of ZnO and C Co-Modification

**DOI:** 10.3390/nano11010012

**Published:** 2020-12-23

**Authors:** Xiaohua Chen, Yong Li, Juan Wang

**Affiliations:** 1State Key Laboratory for Advanced Metals and Materials, University of Science and Technology Beijing, Beijing 100083, China; 2Shaanxi Key Laboratory of Nanomaterials and Nanotechnology, School of Mechanical & Electrical Engineering, Xi’an University of Architecture and Technology, Xi’an 710055, China; xauatli@163.com

**Keywords:** lithium-ion batteries, LiFePO_4_, co-modification, nanomaterials, electron conductivity

## Abstract

Olivine-structure LiFePO_4_ is considered as promising cathode materials for lithium-ion batteries. However, the material always sustains poor electron conductivity, severely hindering its further commercial application. In this work, zinc oxide and carbon co-modified LiFePO_4_ nanomaterials (LFP/C-ZnO) were prepared by an inorganic-based hydrothermal route, which vastly boosts its performance. The sample of LFP/C-xZnO (x = 3 wt%) exhibited well-dispersed spherical particles and remarkable cycling stability (initial discharge capacities of 138.7 mAh/g at 0.1 C, maintained 94.8% of the initial capacity after 50 cycles at 0.1 C). In addition, the cyclic voltammetry (CV) and electrochemical impedance spectroscopy (EIS) disclose the reduced charge transfer resistance from 296 to 102 Ω. These suggest that zinc oxide and carbon modification could effectively minimize charge transfer resistance, improve contact area, and buffer the diffusion barrier, including electron conductivity and the electrochemical property. Our study provides a simple and efficient strategy to design and optimize promising olivine-structural cathodes for lithium-ion batteries.

## 1. Introduction

The ordered olivine lithium-iron phosphate (LiFePO_4_) has been regarded as a power source in various portable electronic devices, such as mobile phones, laptop computers [1,2], digital cameras, and recently, even in electrical vehicles (EVs) and hybrid electric vehicle (HEVs), in that it has fascinating high theoretical capacity (170 mAh/g), structural stability, good safety, and is environment-friendly [3,4]. However, despite these advantages, there are still some severe problems related to this sort of electrode material, which mainly lie in its extremely low electric conductivity (10^−9^ to 10^−10^ S·cm^−1^) at room temperature and slow lithium-ion diffusion velocity, which result in poor electrochemical performance and hide the commercial application of LiFePO_4_ [5,6].

In view of the above existing deficiency of LiFePO_4_ cathode material, there are lots of valid methods that have been presented to optimize the material properties. They include minimizing the particle size of LiFePO_4_ through using a great diversity of systematic methods, such as sol-gel [7,8], solvothermal [9,10,11] and solid-phase methods [12,13] and forming conductive carbon coating on the surface of LiFePO_4_ particles via surfactant-assisted [14,15,16]. In particular, the metal oxide was the first choice for surface modification. Not only could be the impedance of LiFePO_4_ cathode material be reduced, but also it enhanced its electrochemical performance [17]. Liu et al. studied the co-coating of zirconia and carbon on LiFePO_4_ and found that the coating of zirconia decreased the charge transfer impedance and enhanced its rate capability [18]. Park et al. reported vanadium oxide coating on LiFePO_4_ and found that the Coulombic efficiency and the reversibility were improved [19]. Chang et al. found that the high-temperature structural stability of the electrode material was further improved after titanium oxide was coated on LiFePO_4_ [20]. However, most of the metal oxides that were reported for coating, so far, are non-conductive oxides, which could have some impact on the conductivity of LiFePO_4_.

In this work, we proposed LiFePO_4_/C-ZnO composites synthesized via a hydrothermal method. ZnO (0 wt%, 1 wt%, 3 wt%, 5 wt%), as a large bandgap semiconducting material, is directly dispersed in LiFePO_4_ material using ball milling, and provides good electrical contacts at the surface of the active electrode material particle and facilitates fast lithium-ion/electron transfer. Meanwhile, the morphology and electrochemical properties were systematically investigated, the results show that LiFePO_4_ with ZnO and carbon co-modification not only improved the electronic conductivity, but also reduced the diffusion path of lithium-ion.

## 2. Experimental procedure

### 2.1. Sample Synthesis

According to the mole ratio of the proper amount of ferrous sulfate (FeSO_4_·7H_2_O), lithium carbonate (Li_2_CO_3_), and ammonium dihydrogen phosphate (NH_4_H_2_PO_4_) as raw materials were dissolved into the appropriate amount of deionized water, which was taken into the proper amount of oxalic acid. Then, the final solution was transferred to an autoclave and incubated at 180 °C for 12 h in an oven. The cleaned samples were put into the deionized water with glucose and a proper amount of zinc carbonate, to make the carbon and zinc oxide (0 wt%, 1 wt%, 3 wt%, 5 wt%), was evenly coated, hence, we transferred it to the ball mill for 3 h, and dried the sample at 80 °C for 12 h. Then, we put the samples in the tube furnace at 700 °C for 6 h under N_2_ protection. Eventually, LiFePO_4_/C-ZnO powders were obtained.

### 2.2. Structural–Morphological Characterization

The X-ray diffraction (XRD) patterns were obtained on X-ray diffraction (XRD) via using XRD-7000 X-ray diffraction (XRD, Shimadzu Labx XRD-7000, Tokyo, Japan) with Cu-Ka radiation diffractometer, and operating at 40 kV and 40 mA in an angular range of 2*θ* = 10–90° with a scan rate of 8°/min. The morphology of the as-synthesized powder was observed by scanning electron microscopy SEM) (JSM-7610F electron microscope from JEOL, Tokyo, Japan Electronics Co. Ltd) with a voltage of 20 kV. The transmission electron microscopic (TEM) images of the samples were taken on JEM-3010 (JEM-2100, Tokyo, Japan) with an acceleration voltage of 300 kV.

### 2.3. Electrochemical Characterization

The prepared material, acetylene black, and polyvinylidene fluoride (PVDF) were mixed at a weight ratio of 8:1:1 in N-methyl-2-pyrrolidone (NMP) until forming a fully homogeneous slurry. The slurry was coated onto an Al foil substrate. After vacuum drying at 80 °C for 12 h, the electrodes (working electrode) were obtained and transferred to an argon-filled glove box (MBRAUN LABSTAR, Shanghai, China, H_2_O ≤ 0.1 ppm, O_2_ ≤ 0.1 ppm) for cell assembly. Pure lithium wafers served as counter electrodes. Coin cells (CR2032) were used to assemble cells, with Celgard 2400 separator film (diameter: 16 mm; thickness: 20 μm), in which a little electrolyte (1M LiPF_6_ in ethylene carbonate/dimethyl carbonate (1:1, *v*/*v*)) was deposed.

The CR2032 cells were employed to test all the electrochemical performances at room temperature. Cyclic voltammetry (CV) tests were carried out using an electrochemical analyzer (ParStat-2273). The CV measurements were performed between 2.5 and 4.2 V vs. Li/Li^+^ at a scanning rate of 0.5 mV·s^−1^. The charge/discharge tests were operated between 2.5 and 4.2 V using a battery testing system (NewareBTS-4008-5V10mA, neware electronics Co., Ltd. Shenzhen, China). Electrochemical impedance spectroscopy (EIS) measurements were carried out (over a frequency range of 100 kHz to 10 mHz with an applied amplitude of 5 mV) on ParStat-2273.

## 3. Results and Discussion

### 3.1. Structural Analysis

X-ray powder diffraction (XRD) patterns of LiFePO_4_/C-xZnO composites with the different contents of ZnO are shown in Figure 1. The crystal phases of all samples under high-temperature treatment for sufficient time are in accordance with the ordered olivine structure indexed orthorhombic Pnma (JCPDS No.83-2092). The sharp peaks for all the synthesized samples indicate that the samples are well crystalline [21]. In addition, the LiFePO_4_ structure was not affected by the carbon coating in that the carbon diffraction peak was not detected.

### 3.2. Morphological Analysis

The morphologies of as-prepared samples of LiFePO_4_ composites were characterized by SEM, as shown in Figure 2. The figure shows that the LFP/C-ZnO (3 wt%) sample had the best uniformity and dispersibility, and the size was smaller between 400–600 nm, which disclosed that smaller nanoparticles and excellent dispensability (LFP/C-xZnO (x = 3 wt%)) are beneficial for accelerating the electrochemical reaction between the electrolyte and the particle interface, to optimize the electron and ionic conductivity of LiFePO_4_.

In order to clearly confirm the presence of ZnO on the LiFePO_4_, element mapping results of LFP/C-xZnO (3 wt%) composites were displayed, as shown in Figure 3. The element mapping images of a well-dispersed LFP/C-x ZnO (3 wt%) by SEM further verify the uniform distribution of the host Fe, P, and O elements, as well as the homogeneous C/ZnO coating layer elements, i.e., C, Zn, and O. Then, The TEM of ZnO and carbon co-coating was also observed, as shown in Figure 4. In some places, the carbon layer is only 1–2 nm, and even some positive electrode materials are exposed outside the coating layer [22]. However, for the LiFePO_4_ samples coated with carbon and zinc oxide, the coating thickness was 4–5 nm and was very uniform. The additional ZnO repaired the discontinuity and incompleteness of the carbon layer. Such a coating layer can not only improve the conductivity of LiFePO_4_, but also protect the corrosion of LiFePO_4_ by electrolyte during charging and discharging.

### 3.3. Electrochemical Performance

The activated LFP/C-xZnO cathode material was assembled into coin cells to test the electrochemical properties. The initial charge/discharge capacity of all samples, at different contents of ZnO, is shown in Figure 5. The charge/discharge capacity for prepared LFP/C-xZnO with various contents of ZnO (0, 1 wt%, 3 wt%, 5 wt%) were 125.4/123.8 mAh/g, 131.2/129.9 mAh/g, 139.5/138.7 mAh/g, and 133.1/131.1 mAh/g at 0.1 C rate, respectively. Moreover, it can be seen from the figures that the charging platform and discharging platform of all samples are obvious and stable, indicating that these samples have excellent electrochemical properties. With the increase of the contents of ZnO up to 5 wt%, they do not show a continuously increased capacity, but the electrochemical performances of electrodes at the 3 wt% of ZnO concentration are the best. The initial discharge capacity of LFP/C-xZnO (x = 3 wt%) at 0.1C rate is 138.7 mAh/g which is the highest among all the samples. The possible reason was analyzed that the existence of zinc oxide repaired the carbon layer, making it more continuous, complete, and uniform, which can not only promote the conductivity, but also protect the cathode material from the corrosion of electrolyte. However, the charging–discharge ratio of the LFP/C-xZnO (x = 5 wt%) is lower than that of the LFP/C-xZnO (x = 3 wt%), which may be largely attributed to the blocking of lithium-ion transport through a thicker coating layer of ZnO, the layer led to more polarization rather than the electrochemical enhancement from the increased electronic conductivity [23]. Hence, zinc oxide, as an auxiliary, cannot exist too much.

The cyclic performance of the LFP/C-ZnO (x = 3 wt%) composite shows good cycling stability. The discharge capacity retention for different contents of ZnO (0, 1 wt%, 3 wt%, 5 wt%) composites are 105.5, 117.3, 131.5, and 121.3 mAh/g after 50 cycles at 0.1C rate, respectively, as shown in Figure 6. The corresponding capacity retention rates are 85.2%, 90.3%, 94.8%, and 92.4%. It can be seen from the figure that the rates after 50 cycles are more than 90% via the co-coating of zinc oxide and carbon. While the sample with only carbon coating has the lowest capacity retention rate of 85.2%, and it can also be seen from the curve that its stability is the worst. The electrolyte will constantly erode the cathode material and the structure of the material will change as lithium-ion continuous extraction/insertion when the battery is circulating [24]. However, Zinc oxide coating can repair the incompleteness and discontinuity of carbon coating and improve the electrical conductivity and reduce the erosion of electrolyte for cathode materials. Meanwhile, the coating layer modified by zinc oxide becomes more uniform, so that the uniform coating layer can not only prevent the erosion of electrolyte, but also reduce the structural deformation of LiFePO_4_ in the process of lithium-ion continuous extraction/insertion, thereby improving its cycling performance [25].

The electrochemical impedance spectroscopy (EIS) of LFP/C-xZnO (0, 1 wt%, 3 wt%, 5 wt%) electrodes is compared in Figure 7. Generally speaking, there are two distinct parts of all the curves including a depressed semicircle at high frequency and an oblique line at low frequency. The area of high frequency in correspondence with the ohmic resistance (R_e_) reflects the resistance of the electrolyte and electrode, and the charge transfer resistance (R_ct_) refers to the semicircle in the middle-frequency range in the cathode/electrolyte interface. The diffusion of Li-ions in LiFePO_4_ named Warburg impedance (Zw) is pointed out from the low-frequency oblique line and C_et_ indicates the capacitance [26]. The sample without zinc oxide coating exhibits the largest charge transfer impedance, which reached 296 Ω. However, the charge transfer impedance of the samples coated with zinc oxide is generally small. The charge transfer impedance of the samples coated with ZnO (1 wt%, 3 wt%, 5 wt%) is 179 Ω, 102 Ω, and 191 Ω, respectively. The joint coating of zinc oxide and carbon not only make the contact surface between electrodes and electrolytes more uniform, but also make the contact area larger, which improved the charge transfer rate between cathode materials and electrolytes and reduced the charge transfer impedance. In addition, the addition of excessive zinc oxide will agglomerate the materials and affect the contact between electrolyte and electrodes. Thus, the charge transfer impedance, compared with that of 3%, will increase when the coating content reaches 5%.

To further study the effect of zinc oxide and carbon co-coating on the electrochemical properties of LFP/C-xZnO (x = 3w%). Cyclic voltammetry (CV) was carried out to discuss the material of LFP/C-xZnO (x = 3 wt%) by using a scanning speed of 0.1 mV·s^−1^ between 2.0–4.2 V. As illustrated in the CV curve of Figure 8, The difference of redox peak potential between the samples before and after 50 cycles at high rate shows that the coating is effective for protecting the cathode material, and improving the conductivity of the cathode material, reducing the polarization of the battery. The sharp and symmetric redox peak curves of the two groups demonstrated the great reversibility and reaction activity, owing to the lithium-ion diffusion rate and the high electronic conductivity based on the spheroidal nano-structure and the formed carbon network [27].

## 4. Conclusions

A series of samples of ZnO and carbon co-coatings for LiFePO_4_ materials were successfully synthesized via a hydrothermal route. The results show that the capacity and cycling stability have been greatly improved. An integrated and continuous conducting layer was formed on the surface of the LiFePO_4_ particles, as corroborated by the TEM results. Impedance spectroscopy measurements indicate that the interfacial resistance decreased markedly. Additionally, the cyclic voltammograms showed that the polarization is negligible during cycling for the sample co-coated with ZnO and carbon. The electrode of LFP/C-xZnO (x = 3 wt%) showed a greater discharge capacity (138.7 mAh/g) compared with the C-LFP cathode material (123.8 mAh/g) at a discharge rate of 0.1C. The capacity fading of the LFP/C-xZnO (x = 3 wt%) electrode was lowered to be only 5.2% after 50 charge/discharge cycles at 0.1C rate.

## Figures and Tables

**Figure 1 nanomaterials-11-00012-f001:**
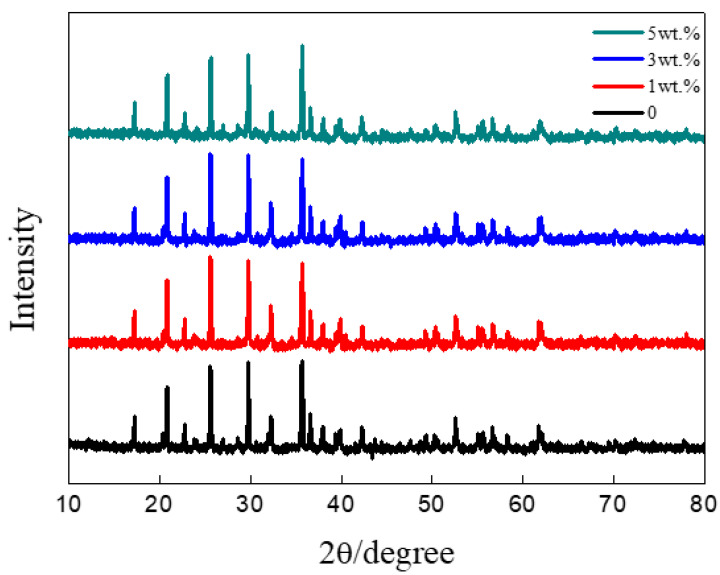
XRD patterns of LiFePO_4_/C-xZnO products with x = 0, 1 wt%, 3 wt%, and 5 wt%.

**Figure 2 nanomaterials-11-00012-f002:**
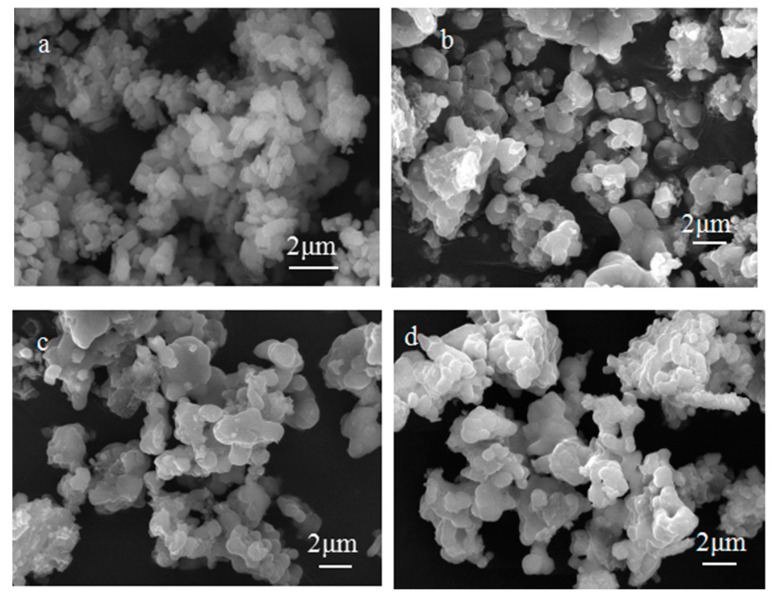
SEM micrographs of LiFePO_4_ composites with different amounts of ZnO: (**a**) 0, (**b**) 1 wt%, (**c**) 3 wt%, and (**d**) 5 wt%.

**Figure 3 nanomaterials-11-00012-f003:**
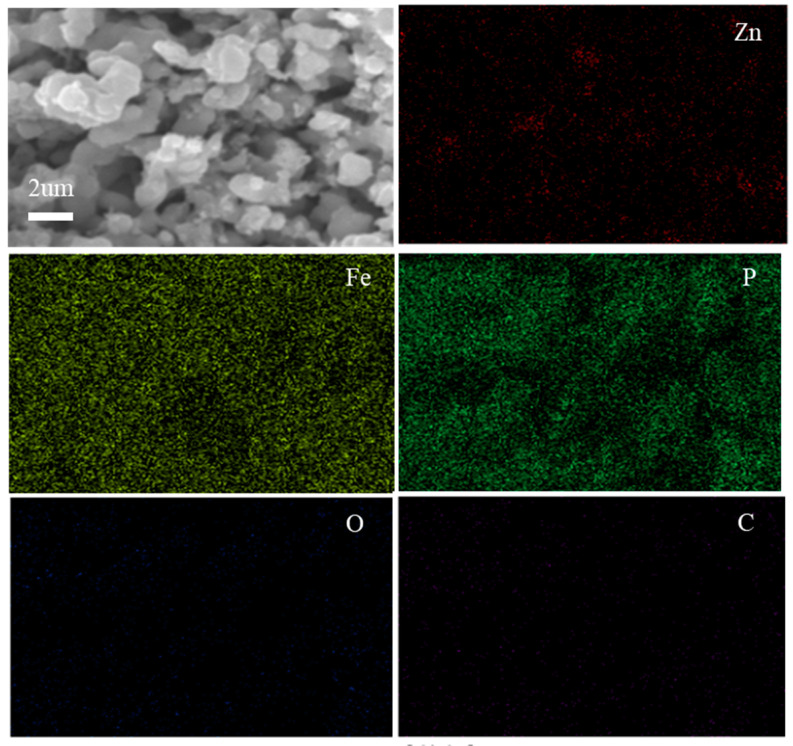
Elemental mapping results of LiFePO_4_/C-xZnO (x = 3 wt%) sample.

**Figure 4 nanomaterials-11-00012-f004:**
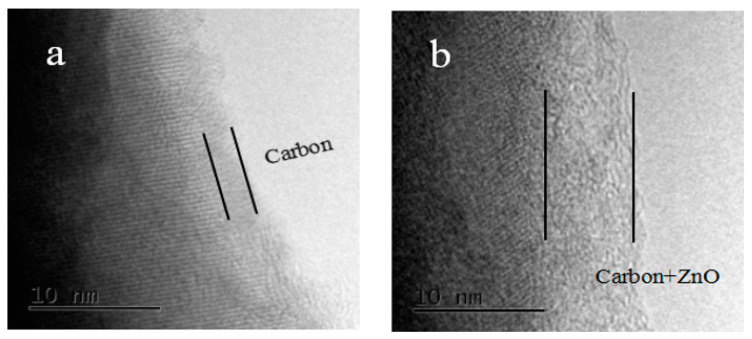
TEM images of LFP/C-xZnO sample of (**a**) bare ZnO doped LiFePO_4_ and (**b**) ZnO of 3 wt% doped LiFePO_4_.

**Figure 5 nanomaterials-11-00012-f005:**
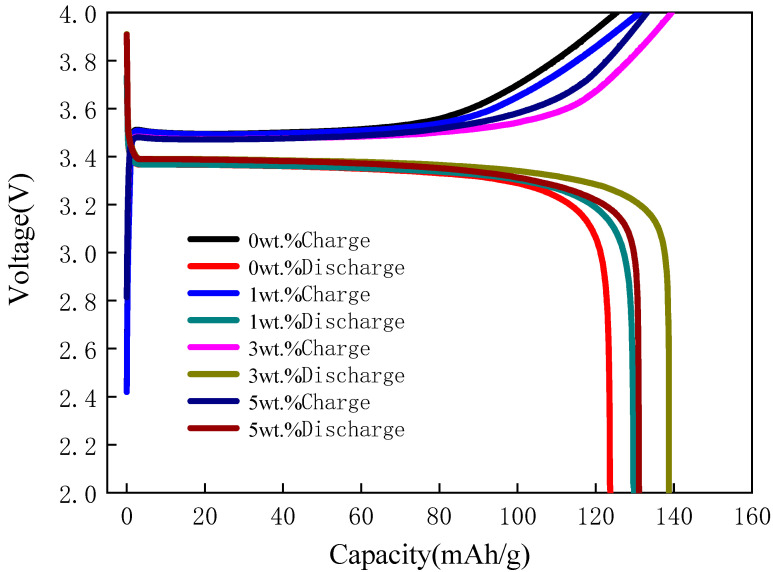
Initial charge/discharge capacity comparison of LFP/C-xZnO (x = 0, 1 wt%, 3 wt%, 5 wt%) at 0.1 C.

**Figure 6 nanomaterials-11-00012-f006:**
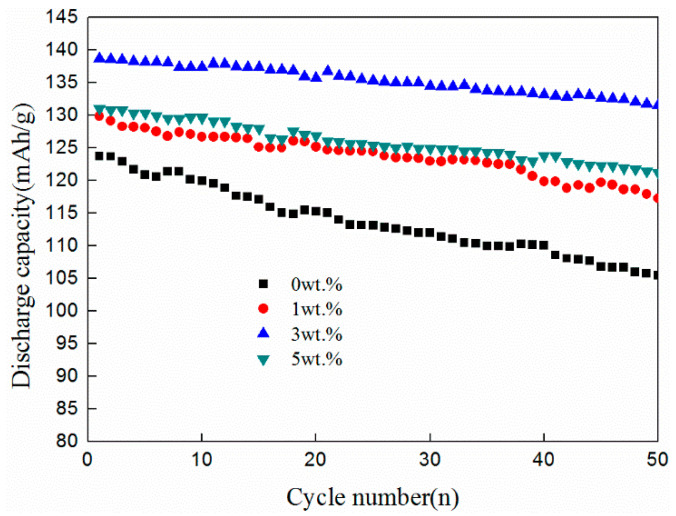
Cycle performance (discharge capacity) comparison of LFP/C-xZnO (x = 0, 1 wt%, 3 wt%, 5 wt%) at 0.1 C.

**Figure 7 nanomaterials-11-00012-f007:**
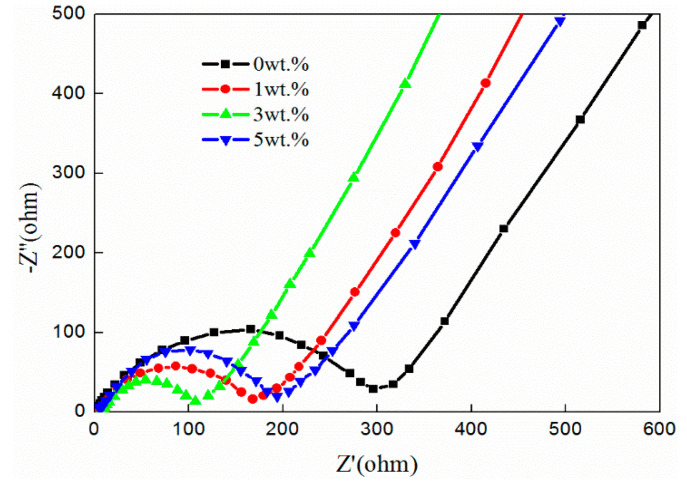
EIS of LFP/C-xZnO (x = 0, 1wt%, 3wt%, 5wt%) samples.

**Figure 8 nanomaterials-11-00012-f008:**
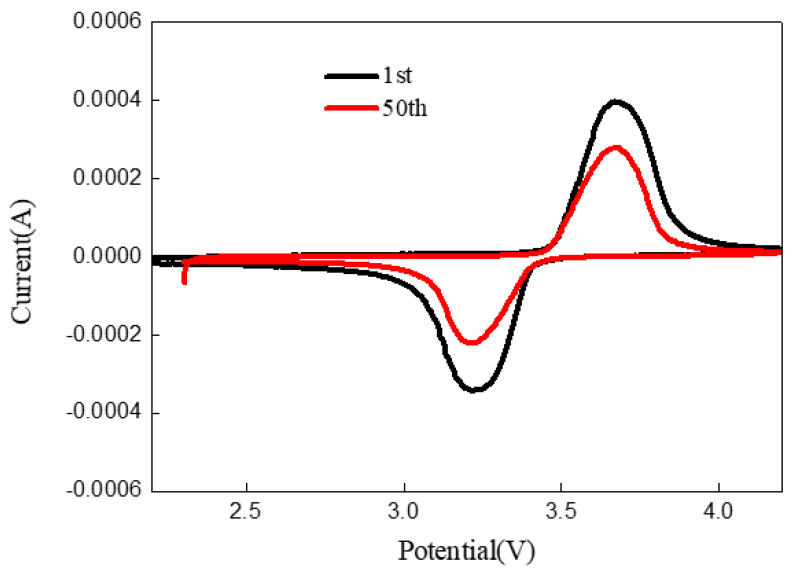
Cyclic voltammetry curves of LFP/C-xZnO (x = 3 wt%) after 1 and 50 cycles.

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
