# Peer review of "Enhanced Electrochemical Performance of LiFePO4 Originating from the Synergistic Effect of ZnO and C Co-Modification"

_nanomaterials, 2020, doi:10.3390/nano11010012_

Round 1
Reviewer 1 Report
The quaility of Tble and all Figures should be improved, also results of SEM, XRD, EDX and other ... should be commented more precise (some aspects, observations were not commented)
the caption for Figure presnted in page 11 has been omitted
It is imposible to determine as well as predict the surface area from SEM micrographs. here should be used other technique.
Also the conclusions are poor in data and, as a result, less informative than the abstract
Author Response
Manuscript Number: ISSN 2079-4991
Title: Enhanced electrochemical performance of LiFePO4 originating from the synergistic effect of ZnO and C co-modification
Dear Editor and Reviewers,
Thank you very much for arduous work and valuable comments. We have studied the comments carefully and the related issues have been well addressed in our revised manuscript according to your suggestions. Revised portion are marked in red and the main corrections in the paper and the responses to the Reviewer’s comments are as flowing:
Comments from Reviewer#1:
Reviewer #1:
1.The quaility of Tble and all Figures should be improved, also results of SEM, XRD, EDX and other ... should be commented more precise (some aspects, observations were not commented)
2.the caption for Figure presnted in page 11 has been omitted
3.It is imposible to determine as well as predict the surface area from SEM micrographs. here should be used other technique.
4.Also the conclusions are poor in data and, as a result, less informative than the abstract
Response: Thank you for your positive and valuable comments. The following concerns have been well addressed accordingly and the revised manuscript is highly expected to be suitable for publishing in Nanomaterials.
- The quaility of Tble and all Figures should be improved, also results of SEM, XRD, EDX and other ... should be commented more precise (some aspects, observations were not commented)
Response: Thank you for your professional comments. We have corrected it in the new manuscript.
- the caption for Figure presented in page 11 has been omitted
Response: Thank you for your professional comments. We have corrected it in the new manuscript.
- It is imposible to determine as well as predict the surface area from SEM micrographs. here should be used other technique.
Response: Thank you for your professional comments. We have corrected it in the new manuscript.
- Also the conclusions are poor in data and, as a result, less informative than the abstract
Response: Thanks for your careful reading of our manuscript. We have corrected it in the new manuscript.
Abstract:
Olivine-structure LiFePO4 is considered as promising cathode materials for lithium-ion batteries. However, the material always sustains poor electron conductivity, severely hindering its further commercial application. In this work, Zinc oxide and carbon co-modified LiFePO4 nanomaterials (LFP/C-ZnO) were prepared by an inorganic-based hydrothermal route, which vastly boosts its performance. The sample of LFP/C-xZnO (x=3 wt%) exhibited well-dispersed spherical particles and remarkable cycling stability (initial discharge capacities of 138.7 mAh/g at 0.1 C, maintained 94.8% of the initial capacity after 50 cycles at 0.1 C). In addition, the cyclic voltammetry (CV) and electrochemical impedance spectroscopy (EIS) disclose the reduced charge transfer resistance from 296 to 102 Ω. These suggest that Zinc oxide and carbon modification could effectively minimize charge transfer resistance, improve contact area, and buffer the diffusion barrier, including electron conductivity and the electrochemical property. Our study provides a simple and efficient strategy to design and optimize promising olivine-structural cathodes for lithium-ion batteries.

Reviewer 2 Report
Wang and colleagues report a nanocomposite composed of ZnO and carbon on LiFePO4. They tested varying amounts of ZnO and used techniques such as XRD, SEM, TEM to analyse the morphology and composition. Although from the data they collected we can infer favourable effects when used in devices, the data in this manuscript needs to be improved vastly before publication and there data collected are not trustworthy and not enough efforts are put into the work. Therefore, I am afraid I have to reject this manuscript. I left some comments for the authors to consider.
The English in this manuscript needs to be improved. There are careless mistakes here and there. For example, from even the title, “ZnO and C co-modified” not “ZnO/C-comodified”. There needs to be a space between parenthesis eg. electrical vehicles(EVs) à electrical vehicles (EVs).
XRD data in Fig.1 needs to be assigned with the indices and the author still need to provide XRD data for ZnO. In addition, C as far as I am concerned does not have any crystallinity, so I am not sure what kind of XRD data the author was expecting from C.
For Figure 2, the authors need to provide a more thorough analysis using image tools such as ImageJ to claim that there is difference in the porosity and surface area. Also, the authors need to provide more SEM data to ascertain the reproducibility. The surplus data can be added to Supporting Information.
The authors need to provide EDX mapping data of all the samples and focus only on the peaks of interest, which in this case is Zn C and etc.
There are many more mistakes and appalling quality in data that I can go on for a long time.
Author Response
Manuscript Number: ISSN 2079-4991
Title: Enhanced electrochemical performance of LiFePO4 originating from the synergistic effect of ZnO and C co-modification
Dear Editor and Reviewers,
Thank you very much for arduous work and valuable comments. We have studied the comments carefully and the related issues have been well addressed in our revised manuscript according to your suggestions. Revised portion are marked in red and the main corrections in the paper and the responses to the Reviewer’s comments are as flowing:
Comments from Reviewer#2:
Reviewer #2: Wang and colleagues report a nanocomposite composed of ZnO and carbon on LiFeP3O4. They tested varying amounts of ZnO and used techniques such as XRD, SEM, TEM to analyse the morphology and composition. Although from the data they collected we can infer favourable effects when used in devices, the data in this manuscript needs to be improved vastly before publication and there data collected are not trustworthy and not enough efforts are put into the work. Therefore, I am afraid I have to reject this manuscript. I left some comments for the authors to consider.
Response: Thank you very much for arduous work and valuable comments. We have studied the comments carefully and the following revision concerns have been well addressed accordingly. We appreciate your valuable and professional suggestions to improve the quality of our manuscript and the revised manuscript is highly expected to be suitable for publishing in Nanomaterials .
- The English in this manuscript needs to be improved. There are careless mistakes here and there. For example, from even the title, “ZnO and C co-modified” not “ZnO/C-comodified”. There needs to be a space between parenthesis eg. electrical vehicles(EVs) à electrical vehicles (EVs).
Response: Thank you very much for arduous work and valuable comments. The corrected results have been well exhibited accordingly.
For example, Title: “Enhanced electrochemical performance of LiFePO4 originating from the synergistic effect of ZnO and C co-modification”.
In this work, Zinc oxide and carbon co-modified LiFePO4 nanomaterials (LFP/C-ZnO) were prepared by an inorganic-based hydrothermal route, which vastly boosts its performance.
- XRD data in Fig.1 needs to be assigned with the indices and the author still need to provide XRD data for ZnO. In addition, C as far as I am concerned does not have any crystallinity, so I am not sure what kind of XRD data the author was expecting from C.
Response: Thank you for your professional comments. We have corrected it in the new manuscript. X-ray powder diffraction (XRD) patterns of LiFePO4/C-xZnO composites with the different contents of ZnO are shown in Fig 1. The crystal phases of the all samples under high temperature treatment for sufficient time are in accordance with the ordered olivine structure indexed orthorhombic Pnma (JCPDS No.83-2092). The sharp peaks for all the synthesized samples indicate that the samples are well crystalline [21]. In addition, the LiFePO4 structure was not affected by the carbon coating in that the carbon diffraction peak did not detect.
Fig. 1 XRD patterns of LiFePO4/C-xZnO products with x= 0, 1wt %, 3wt %, 5wt %.
- For Figure 2, the authors need to provide a more thorough analysis using image tools such as ImageJ to claim that there is difference in the porosity and surface area. Also, the authors need to provide more SEM data to ascertain the reproducibility. The surplus data can be added to Supporting Information.
Response: Thank you for your professional comments. We have corrected it in the new manuscript.
- The authors need to provide EDX mapping data of all the samples and focus only on the peaks of interest, which in this case is Zn C and etc.
Response: Thank you for your professional comments. We have corrected it in the new manuscript.
- There are many more mistakes and appalling quality in data that I can go on for a long time.
Response: Thank you for your professional comments. We have corrected it in the new manuscript.
We appreciate your consideration of our revised manuscript for publication in Nanomaterials. We hope that the revisions in the manuscript and our responses will be sufficient to make our manuscript suitable for its publication.
Once again, thank you very much for your suggestions and warm work. If you have any queries, please don’t hesitate to contact me at the address below.
Thank you and best regards.
Yours sincerely,
Xiaohua Chen
Corresponding author: Xiaohua Chen
E-mail: chenxiaohua710103@126.com

Round 2
Reviewer 1 Report
the manuscript has been significantly improved and after minor linguistic and letter corrections it can be published
Author Response
In order to clearly confirm the presence of ZnO on the LiFePO4, element mapping results of LFP/C-xZnO (3wt%) composites were displayed in Fig 3. The element mapping images of a well-dispersed LFP/C-x ZnO (3wt%) by SEM further verify the uniform distribution of the host Fe, P, O elements, as well as the homogeneous C/ZnO coating layer elements, i.e., C, Zn, and O. Then, The TEM of ZnO and carbon co-coating was also observed in Fig 4. In some places, the carbon layer is only 1nm-2nm, and even some positive electrode materials are exposed outside the coating layer [22]. However, for the LiFePO4 samples coated with carbon and zinc oxide, the coating thickness was 4mn-5nm and was very uniform. The additional ZnO repaired the discontinuity and incompleteness of the carbon layer. Such a coating layer can not only improve the conductivity of LiFePO4, but also protect the corrosion of LiFePO4 by electrolyte during charging and discharging.
